# Population health impact and economic evaluation of the CARDIO4Cities approach to improve urban hypertension management

**Theresa Reiker**[1][�]*, **Sarah Des Rosiers**[1☉], **Johannes Boch**[1], **Gautam Partha**[2], **Lakshmi Venkitachalam**[3], **Adela Santana**[3], **Abhinav Srivasatava**[2], **Joseph Barboza**[4], **Enkhtuya Byambasuren**[5], **Yara C. Baxter**[6], **Karina Mauro Dib**[7], **Naranjargal Dashdorj**[8], **Malick Anne**[9], **Renato W. de Oliveira**[10], **Mariana Silveira**[11], **Jose M. E. Ferrer**[3], **Louise Morgan**[3], **Olivia Jones**[3], **Tumurbaatar Luvsansambuu**[12], **Luiz Aparecido Bortolotto**[13,14], **Luciano Drager**[13,15,16], **Alvaro Avezum**[16,17], **Ann Aerts**[1]

1 Novartis Foundation, Basel, Switzerland, 2 Novartis Healthcare Pvt. Ltd., Hyderabad, India, 3 American Heart Association, Dallas, Texas, United States of America, 4 Intrahealth, Dakar, Senegal, 5 Mongolian Public Health Professionals' Association, Ulaanbaatar, Mongolia, 6 YC Baxter, São Paulo, Brazil, 7 Secretaria Municipal da Saúde, São Paulo, Brazil, 8 Onom Foundation, Ulaanbaatar, Mongolia, 9 Ministère de la Santé et de l'Action Sociale, Dakar, Senegal, 10 IQVIA, São Paulo, Brazil, 11 Instituto Tellus, São Paulo, Brazil, 12 Capital City Health Department, Ulaanbaatar, Mongolia, 13 Instituto do Coração (InCor), Hospital das Clinicas da Faculdade de Medicina da Brazil, São Paulo, Brazil, 14 Sociedade Brasileira de Hipertensão, São Paulo, Brazil, 15 Centro de Cardiologia, Hospital Sírio Libanês, São Paulo, Brasil, 16 Sociedade de Cardiologia do Estado de São Paulo (SOCESP), São Paulo, Brazil, 17 Hospital Alemão Oswaldo Cruz, São Paulo, Brazil

☉ These authors contributed equally to this work.
* theresa.reiker@novartis.com

**Data Availability Statement:** The full model including all input data and generated results is contained in the supplementary material.

## Abstract

Cardiovascular disease (CVD) is the leading cause of mortality worldwide, with 80% of that mortality occurring in low- and middle-income countries. Hypertension, its primary risk factor, can be effectively addressed through multisectoral, multi-intervention initiatives. However, evidence for the population-level impact on cardiovascular (CV) event rates and mortality, and the cost-effectiveness of such initiatives is scarce as long-term longitudinal data is often lacking. Here, we model the long-term population health impact and cost-effectiveness of a multisectoral urban population health initiative designed to reduce hypertension, conducted in Ulaanbaatar (Mongolia), Dakar (Senegal), and in the district of Itaquera in São Paulo (Brazil) in collaboration with the local governments. We based our analysis on cohort-level data among hypertensive patients on treatment and control rates from a real-world effectiveness study of the CARDIO4Cities approach (built on quality of care, early access, policy reform, data and digital, Intersectoral collaboration, and local ownership). We built a decision tree model to estimate the CV event rates during implementation (1–2 years) and a Markov model to project health outcomes over 10 years. We estimated the number of CV events averted and quality-adjusted life-years gained (QALYs through the initiative and assessed its cost-effectiveness based on the costs reported by the funder using the incremental cost effectiveness ratio (ICER) and published thresholds. A one-way sensitivity analysis was performed to assess the robustness of the results. The modelled patient cohorts included 10,075 patients treated for hypertension in Ulaanbaatar, 5,236 in Dakar, and 5,844

**Funding:** The Novartis Foundation was the funder of the initiative, oversaw its implementation and data collection, designed the study, planned the publication and prepared the manuscript. Authors SDR, TR, JBo and AAe are employed by Novartis Foundation. Novartis Foundation further provided direct funding to IntraHealth International (author: JBa), Instituto Tellus (MS), Onom Foundation (ND) and the Mongolian Public Health Professionals' Association, MPHPA (EB) to implement the activities of the CARDIO4Cities approach. YC Baxter (YB) received direct funding for consulting and coordinating the CARDIO4Cities activities in São Paulo on behalf of Novartis Foundation. IQVIA (RWO) received funding to collect data on the hypertension care cascade in São Paulo. Novartis Healthcare Pvt. Ltd., Hyderabad, India (GP and ASr) received funding to conduct the economic analysis. The American Heart Association (LV, ASa, JF and LM) was a recipient of funding from the Novartis Foundation for technical assistance and resources to support professional education, quality improvement, monitoring, and evaluation.

**Competing interests:** I have read the journal's policy and the authors of this manuscript have the following competing interests: TR, SDR, JBo, and AAe are directly employed by the Novartis Foundation. GP, LV, ASa, ASr, JBa, EB, YB, ND, RWO, MS, JF, LM and OJ have directly or indirectly through their employer received funding from Novartis Foundation.

in São Paulo. We estimated that 3.3–12.8% of strokes and 3.0–12.0% of coronary heart disease (CHD) events were averted during 1–2 years of implementation in the three cities. We estimated that over the subsequent 10 years, 3.6–9.9% of strokes, 2.8–7.8% of CHD events, and 2.7–7.9% of premature deaths would be averted. The estimated ICER was USD 748 QALY gained in Ulaanbaatar, USD 3091 in Dakar, and USD 784 in São Paulo. With that, the intervention was estimated to be cost-effective in Ulaanbaatar and São Paulo. For Dakar, cost-effectiveness was met under WHO-CHOICE standards, but not under more conservative standards adjusted for purchasing power parity (PPP) and opportunity costs. The findings were robust to the sensitivity analysis. Our results provide evidence that the favorable impact of multisector systemic interventions designed to reduce the hypertension burden extend to long-term population-level CV health outcomes and are likely cost-effective. The CARDIO4Cities approach is predicted to be a cost-effective solution to alleviate the growing CVD burden in cities across the world.

## Introduction

Cardiovascular disease (CVD) continues to be the leading cause of death worldwide [1]. Over 80% of related deaths occur in low- and middle-income countries (LMICs), where CVD often arises earlier in life and with worse outcomes [2]. In many LMICs, CVDs have replaced infectious diseases as the number one cause of mortality [1]. Air pollution, the spread of unhealthy diet and lifestyle choices, such as smoking and physical inactivity, and rapid urbanization contribute to a rapidly growing CVD burden, alongside widening health inequities and shortage of adequate health infrastructure and professionals [3,4]. Countries such as Brazil, Senegal and Mongolia report 20–60% increases in strokes and coronary heart disease (CHD) over the past 10 years [5]. Hypertension or high blood pressure (BP) is the prime risk factor for CVD and accounts for over half of the stroke and CHD burden, causing 40% of stroke-related mortality [6,7]. Over 1 billion people with hypertension (82% of the global total) live in LMICs [8], with half of them unaware of the condition and less than 10% achieving BP control [8–10]. The additional strain exerted on health systems by the COVID-19 pandemic further compromises cardiovascular and hypertension testing and treatment [11].

In addition to population health consequences, CVD poses a substantial economic burden. In 2010, it was estimated that CVD would lead to total economic losses of USD 3.76 trillion across LMICs between 2011 and 2025, representing 2% of their 2010 joint Gross Domestic Product [12,13]. The cost of CVD is expected to grow as the prevalence of risk factors increases [8]. Based on financial trends observed in high-income countries, e.g., the doubling of CVD costs in the United States between 1995 and 2016 [14], the cost of CVD in LMICs can be expected to exceed old estimates, threatening sustainable economic growth.

A primary strategy for improving cardiovascular (CV) population health in LMICs is improving hypertension care [15–17]. Addressing hypertension at primary care level, including non-medication interventions, is one of the most cost-effective methods for minimizing its public health burden [18–20]. Hypertension is easy to diagnose, and reductions as small as 10 mmHg systolic BP or 5 mmHg in diastolic BP can significantly decrease the risk of stroke and myocardial infarction [21]. As financial resources in LMICs are limited, demonstrating the impact and cost-effectiveness of population-level interventions is important. International guidelines promote individual low-cost, scalable interventions targeting hypertension [22,23]. However, effectively and sustainably improving CV population health in LMICs requires

systemic improvements [24] and the economics of such intervention packages in LMICs remain understudied [25,26]. Only 105 studies reported on non-medication public health interventions addressing hypertension in LMICs globally, between 2007 and 2017 (49 of which simultaneously targeted diabetes) [18]. The majority neither contained replicable methods, nor reported on the effectiveness or cost-implications of the approach [18].

In 2018, a global multisector urban health initiative was launched in three low-and middle-income settings, aiming to reduce the hypertension burden and improve cardiovascular population health. The initiative implemented the CARDIO4Cities approach, shorthand for quality of Care, early Access, policy Reform, Data and digital, Intersectoral collaboration, and local Ownership [27,28]. It was piloted in Ulaanbaatar, Mongolia, Dakar, Senegal, and in the district of Itaquera in São Paulo, Brazil, and offered a broad portfolio of health intervention options. Based on the local needs around hypertension care, city health authorities and the initiative's local implementing partners co-developed tailored intervention packages to address bottlenecks in the CV population health roadmaps. Interventions included: standardized hypertension management for frontline health workers with simplified care algorithms and clinical decision support tools, systematic early detection through BP measurement within health facilities and at high traffic venues throughout the cities (such as subway stations or samba schools), and establishing data collection for monitoring progress, evaluating outcomes, and supporting data-driven decision-making. Specific interventions in each city are summarized in S1 Table. Following less than two years implementation of the CARDIO4Cities approach, BP control rates among patients in primary health centers increased from 12% to 31% in São Paulo, from 7% to 19% in Dakar, and from 3% to 19% in Ulaanbaatar [29].

Here, we expand from those previously reported health improvements and provide an evaluation of the CARDIO4Cities approach's population health and economic impact. In this study, we estimate the number of CV events (CHD and stroke) averted over time and evaluate the approach's cost-effectiveness across three geographies.

## Methods

### Data collection and study population

Details on the approach of the urban population health initiative and its outcomes on hypertension care have been reported previously [28,29]. Following needs assessments and stakeholder alignments locally, data collection began at different time points in the three cities. In Ulaanbaatar, it began in quarter 1 (Q1) 2018. For São Paulo, the collaborative roadmap design for implementation with local authorities and clinics started in December 2017, and the solutions of the CARDIO4Cities approach were rolled out in Q4 2018. In this study, we thus consider Q4 2018 as the start of the intervention period to evaluate the cost-effectiveness of the solutions. In Dakar, implementation was initiated at different times in the different districts: in the West, implementation started in Q2 2018, in the Center and North in Q3 2018, and in the South in Q1 2019. This resulted in reporting periods of 21, 19, and 15 months in Ulaanbaatar, Dakar, and São Paulo, respectively. Overall, the CARDIO community events reached an estimated 1.2 million people in Ulaanbaatar, 1.3 million in Dakar, and 1.0 million in São Paulo.

Data on the total number of patients diagnosed, treated, and controlled for hypertension were extracted quarterly from patient medical records of 23 participating clinics in Ulaanbaatar, and 66 in Dakar. In São Paulo, data was available in a sample of 6 out of 24 primary health centers of the Itaquera district. The centers were selected by the São Paulo Secretary of Health to ensure that all six health center management models the city were represented. In São Paulo, data was only available for patients who had previously provided written consent. Demographic data were available on patients' sex, age, and BP. Further, patients' CV risk was

**Table 1. Patient characteristics and outcomes.** This table summarizes previously published population characteristics, intervention periods, and differences in hypertension control rates following the implementation of the CARDIO4Cities approach in Ulaanbaatar, Dakar, and São Paulo.

| | Ulaanbaatar | Dakar | São Paulo |
|---|---|---|---|
| Intervention period | Q1 2018 –Q3 2019 | Q1 2019 (Q2 2018)–Q4 2019* | Q4 2018 –Q4 2019 |
| Health Services Catchment Area | 1.2 million | 1.3 million | 1.0 million |
| Number of patients treated for hypertension** | 10,075 | 5,236 | 5,844*** |
| Mean age | 61 | 58 | 62 |
| Hypertension control threshold | ≥ 130/80 mmHg | > = 140/90 mmHg | > = 140/90 mmHg |
| BP control rate at baseline | 3% | 13% (7%) * | 12% |
| BP control rate following CARDIO implementation | 19% | 19% | 31% |

* The baseline control rate in the West Dakar (start Q2 2018) was 7%. By the time the whole city was included (Q1 2019), the control rate was 13%.

** Total number of patients treated for hypertension were extracted from published data [29].

*** In São Paulo, CARDIO interventions covered all 24 primary health centers of the Itaquera district, but data was collected in a random sample of 6 centers. The number of total patients treated for hypertension in the 6 centers (n = 1461) was therefore multiplied by 4 to extrapolate to the total cohort.

evaluated by physicians during their initial medical assessment, and categorized as low, medium, or high (see [29] S1 Table), implicitly capturing the effects of age, risk behaviors, and comorbidities. Systolic BP was collected longitudinally from de-identified primary health records of patients ≥18 years old. Patient and relevant population characteristics have been described previously [29] and are summarized in Table 1. The definition of hypertension followed national guidelines: in Dakar and São Paulo, a BP threshold of ≥140/90 mmHg was used, in Ulaanbaatar hypertension was defined as BP ≥ 130/80 mmHg. Based on patients' BP values at their first reported visit, previously diagnosed patients were classified as controlled or uncontrolled for hypertension, and newly diagnosed patients as uncontrolled. Anti-hypertensive therapy was prescribed according to available national guidelines [29]. For patients treated with medication who had at least two reported consultations with BP measurement within the study period [29], the last reported BP value was used to evaluate whether hypertension control was achieved.

## Estimating CV event rates during the reporting period

We first developed a decision tree model (Fig 1A) to estimate and compare CV event and mortality rates during the reporting period for the treated patient populations in Ulaanbaatar, Dakar, and São Paulo. For each city, we modeled these outcomes for the presence of CARDIO interventions compared to their absence.

A description of the modelling methodology is provided in S1 Text, and model assumptions and parameters are listed in Table D in S1 Text. In brief, total CV event rates were estimated based on the population sex ratio (percentage of male patients in Ulaanbaatar: 37%, Dakar: 22%, São Paulo: 27% [29]), distribution across of CV risk and BP categories, and hypertension control rates. For the 'CARDIO' scenario, CVD risk, BP levels and hypertension control rates were set to those measured among treated patients at the end of the implementation period. For the 'No CARDIO' scenario, they were assumed equal to the first quarter of implementation, Q1 2018 in Ulaanbaatar, and Q4 2018 in São Paulo. For Dakar, while implementation was initiated in Q2 2018, only 360 treated patients were initially included, and CV risk and sex were not consistently recorded. Additionally, only 83 of these patients had their control status evaluated in a follow-up visit. Due to the small and potentially biased sample, we chose Q1 2019 (when expansion to the whole city was reached) with its associated control rate and patient risk distribution as baseline for Dakar [29]. To avoid underestimating the cost-

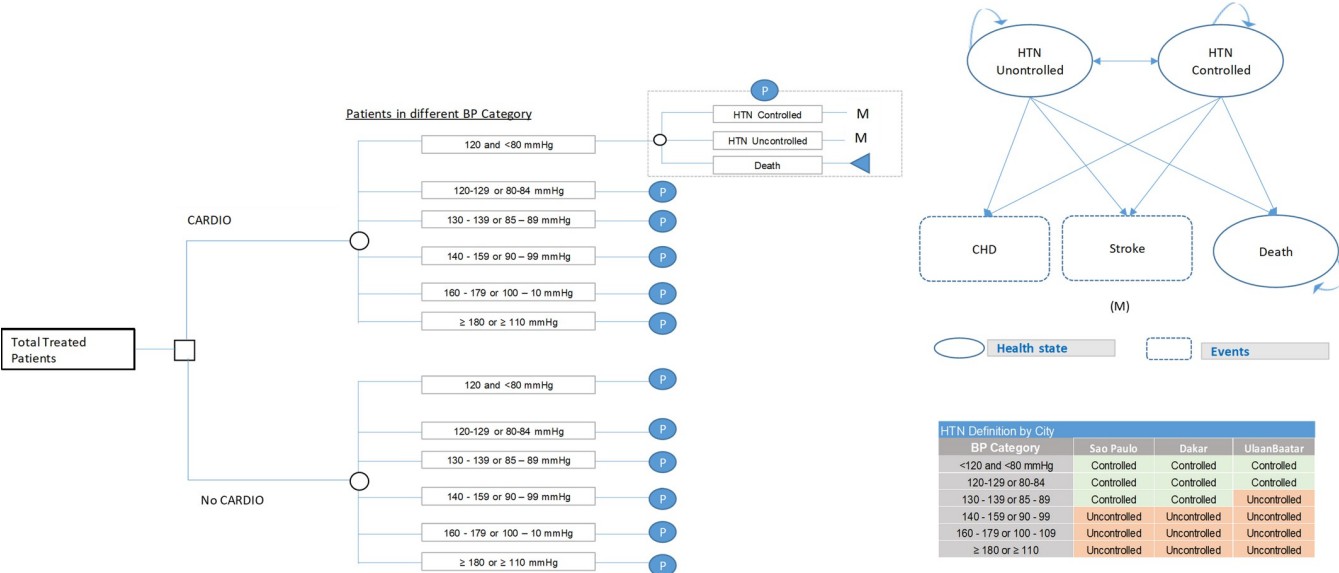

**Fig 1. Schematic overview of the decision tree model for estimating CV events and deaths averted during the CARDIO implementation phase and of the Markov model used for a 10-year projection of population-level CV event rates. A.** Decision tree model. The decision tree model was used to estimate CV events and deaths averted during the implementation phase. Patients were either included in the program (factual) or not (counterfactual). Within each arm, patients were assumed to fall within different blood pressure categories, according to a distribution matching the observed. Based on the BP, patients could achieve hypertension control or not, or die at fixed probabilities. **B.** Markov model. The Markov model assumes three states for patients: Controlled (BP controlled) or uncontrolled hypertension (BP uncontrolled), or death. The distribution of patients across the categories (M) at the beginning of the 10-year projection period was matched to the endpoint of the two-year implementation period (as estimated using the decision tree model). The model allowed for individual patients to remain in a state, transition between controlled or uncontrolled hypertension, or from either state to death. Controlled and uncontrolled patients could experience two health events (CHD event or stroke) at given probabilities.

effectiveness of the initiative in Dakar by disregarding its effect during the expansion period, an additional analysis considering the Q2 2018 control rate and Q1 2019 patient distribution is provided in the (Tables A and B in S3 Text).

Only patients with complete information were included (see S1 Text). The sex-adjusted risk 10-year CV risk for patients with uncontrolled hypertension in each risk category was quantified using the Framingham Risk Score (FRS) [30] (Table B in S1 Text). Based on the number of individuals in each risk category and the sex ratio, baseline CVD event rates at patient population-level were calculated. For the proportion of the patient population with controlled hypertension, relative risk reductions in CVD event rates were applied, based on hazard ratios derived from the literature. Hazard ratios were stratified by sex and initial BP (at first measurement, before control) and are summarized in Table C in S1 Text. In the absence of country-specific rates, it was assumed that 30% of CVD events would be strokes [31]. Outcomes were modelled over two years. The resulting populations were used as baseline for the subsequent extended time model.

## Ten-year projection

Due to the chronic nature of CVD and its associated morbidity, we projected a long-term effect of the initiative into the future, affecting CHD, stroke, and mortality rates after cessation of CARDIO activities. Patient health outcomes for the CARDIO and No CARDIO scenarios were modelled over a 10-year projection phase using a Markov model with a cycle length of one year (Fig 1B). All patients with controlled or uncontrolled hypertension at the end of the reporting period were included for the projection model.

We built a Markov model with three states: controlled, uncontrolled, and deceased. Patients were assigned to start in the state reflecting their BP control status at the last reported measurement in the data (factual) or baseline (counterfactual), assuming that no further programmatic benefits from the initiative would be realized after the reporting period. Controlled and uncontrolled patients were assumed to be at risk of developing a CHD event or stroke or transitioning to death. Age-adjusted, mortality rates by hypertension control status were estimated based on the WHO country life tables (https://www.who.int/data/gho/data/indicators/indicator-details/GHO/gho-ghe-life-tables-by-country). Compared to controlled patients, uncontrolled hypertensive patients had an approximately threefold increased risk of death in uncontrolled patients [32] (see also Table D in S1 Text). CVD event rates for individuals with uncontrolled and controlled hypertension were used as in the decision tree model.

## Cost and cost-effectiveness estimation

**Implementation and health costs.** The assumed costs are summarized in Table 2 (see also S2 Text and Tables A and B in S2 Text for further details). Implementation costs of the initiative included operational budgets for the interventions and local capability strengthening, as well as the data collection but excluded initial capital investments (groundwork activities and ecosystem building) to build the partnership that would implement the CARDIO4Cities approach. The groundwork phase consisted of establishing contact and agreements with government entities, feasibility analysis, stakeholder engagement and first technical workshops, and tender processes to identify the in-country implementation partner to support the initiative. Ecosystem building activities throughout implementation included stakeholder engagement, the set-up of a global quality of care framework and a global shared measurement and evaluation system. The collaboration agreements with the local governments established the funder as a supporting entity to implement the CARDIO4Cities approach under the guidance of the local governments and to complement the already existing capacity and current spending. Annual implementation costs for the funder were USD 663,771 for Ulaanbaatar, USD 428,927 for Dakar, and USD 591,229 for São Paulo.

Estimates for health system costs for each health state and for events were extracted from the literature for São Paulo. An annual baseline health cost of USD 191 was assumed for both uncontrolled and controlled hypertension patients [33] and event costs were estimated at USD 1,522 for CHD [34] and USD 5,864 for stroke [35,36]. Additional costs resulting from productivity loss through partial or total absence from work were estimated from the proportional per-capita income earned in one working day [33]. Costs from productivity loss were adjusted for inflation (2003–2019) using World Bank data [37]. For Dakar and Ulaanbaatar, health system and productivity loss costs were estimated from the São Paulo values, using purchasing power parity (PPP) conversion factors [38].

**Table 2. Cost assumptions.** This table summarizes the cost assumptions made for Ulaanbaatar, Dakar, and São Paulo. PPP = Purchasing power parity.

|  | Ulaanbaatar | Dakar | São Paulo |
|---|---|---|---|
| **PPP conversion factor** | 5.57 | 1.37 | NA |
| **Hypertension cost (in USD)** | 34 | 139 | 191 |
| **CHD event cost (in USD)** | 273 | 1,111 | 1,522 |
| **Stroke event cost (in USD)** | 1,053 | 4,280 | 5,864 |
| **Indirect costs: annual loss of productivity (in USD)** | 27 | 112 | 153 |
| **Net program cost (in USD)** | 663,771 | 428,927 | 591,229 |

**Cost-effectiveness.**   The initiative's cost-effectiveness and associated cost savings compared to usual care were evaluated using incremental cost effectiveness ratios (ICER) and the incremental cost of implementing the CARDIO4Cities approach per quality-adjusted life year (QALY). QALYs were calculated by assuming a reduced quality-of life with increasing age for hypertension- and CVD-related morbidity and mortality. Age-adjusted utility estimates for hypertensive patients and annual utility costs for CHD and stroke events for São Paulo were extracted from the literature and are summarized in Table 3 [39]. In Ulaanbaatar and Dakar, utility weights were assumed to be the same as in São Paulo, due to the lack of EQ-5D data for the hypertension population in those cities.

ICER thresholds adjusted for PPP and in-country opportunity costs were extracted from literature [40]. To reflect the current discourse on defining adequate ICER thresholds, we also considered the WHO-CHOICE- stipulated ICER thresholds of three times the national per-capita GDP. ICER values equal or lower than the ICER thresholds indicated that the CARDIO interventions were cost-effective and considered a likely efficient use of resources for governments and society.

The cumulative number of CHD, stroke events and deaths averted were derived using the 10-year Markov model. The numbers were subsequently used to estimate cumulative cost savings and QALYs gained through the CARDIO implementation. Reduced incidence of CV events and death were converted to QALYs using utility weights to reflect improved quality of life for patients achieving BP control (Table 3). Differences in productivity costs between controlled and uncontrolled hypertension patients were included and analysis carried out from a societal perspective. Annual discounting rates were applied to both cost and benefit (QALY) estimates. For São Paulo, the discounting rate was 5% [41] and for the other two cities 3% based on the WHO recommendation for countries where no standard metric exists [42].

For São Paulo, a cost-effectiveness threshold of USD 3,210 to USD 10,122 per QALY gained was extracted from the literature [40]. Using the purchase power parity (PPP) ratio, thresholds of USD 73–1,166 per QALY gained were derived for Dakar and of 1,624–4,849 per QALY for Ulaanbaatar.

## Sensitivity analysis

A univariate sensitivity analysis was conducted by varying one parameter at a time by 10% and estimating the change in cost-effectiveness. The following parameters were varied to identify which had the greatest effect on total implementation cost: total number of people diagnosed at last time point, direct costs, running costs after implementation, costs of uncontrolled and controlled hypertension, per-event costs of CHD, per-event costs of stroke, annual loss of productivity, utility, and disutility.

**Table 3.  Disutility assumptions for age-adjusted hypertension and annual utility costs for CV events.**
CV = cardiovascular.

| Age-specific hypertension utility | |
|---|---|
| **Age 50–59** | 0.84 |
| **Age 60–69** | 0.82 |
| **Age 70–79** | 0.78 |
| **Age 80–100** | 0.74 |
| **CV event utility cost (annual)** | |
| **Coronary heart disease** | 0.018 |
| **Stroke** | 0.048 |

## Software

The decision tree and Markov models were developed in Microsoft Excel 2016. The full models are provided as a in S1 File.

## Research ethics approval, data availability, and patient consent

In this paper we conduct a secondary analysis, referring to publicly available summary statistics on confidential primary data of de-identified patient records [29]. Approvals for primary data collection were obtained from government entities and/or local ethics committees. The ethics approvals for the respective countries are Letter No. 1/158 dated February 21st, 2018, from the Capital City department of health for Ulaanbaatar, SEN 18/79 and SEN19/14 for Dakar, and CEP-SMS; 3·818·858 for São Paulo with the latter requiring written patient consent. In Dakar and Ulaanbaatar, written consent was not required. In Ulaanbaatar, approval for data collection was granted by the Ulaanbaatar health department.

# Results

## Event rates and cumulative number of events during the reporting period

The average patient-level annual risk of developing a CVD event under the No CARDIO scenario was estimated at 2.29% within the São Paulo cohort, 1.96% in Dakar and 1.08% in Ulaanbaatar. For the CARDIO scenario, these risks were estimated at 2.22%, 1.94%, and 1.07%, respectively.

We estimated that following CARDIO implementation, 10.3% of CHD events and 9.6% of strokes were averted compared to usual care in the patient population of Ulaanbaatar, 3.0%, and 3.3% in Dakar, and 12.0% and 12.8% in São Paulo. We estimated no effect on deaths during the implementation phase.

## Ten-year projection of long-term effects

We estimated that over the subsequent ten years, 3.8% of CHD events and 5.4% of strokes would be averted in the Ulaanbaatar patient group (3.3 CHD events and 1.4 strokes per 1,000 patients). Over the same period in Dakar, we estimated that 2.8% of CHD events and 3.7% of strokes would be averted (4.8 CHD events and 1.9 strokes per 1,000 patients) and in São Paulo, 7.8% of CHD events and 9.9% of strokes (13.9 CHD events and 5.3 strokes per 1,000 patients). We further estimated that 6.9%, 2.7%, and 7.9% of deaths would be averted in Ulaanbaatar, Dakar, and São Paulo, respectively. Table 4 summarizes the expected cumulative number of CHD events, strokes, and deaths after the reporting period and 10 years after the end of the initiative under the 'CARDIO' and 'No CARDIO' scenario. Fig 2 shows the estimated cumulative number of CV events and deaths averted by year throughout the 10-year projection phase.

**Cost-effectiveness analysis.**   Over 10 years, we estimated an additional 884 QALYs (88 QALYs per 1,000 patients) gained in Ulaanbaatar, 121 (23 QALYs per 1,000 patients) in Dakar, and 572 (98 QALYs per 1,000 patients) in São Paulo. The projected cumulative incremental QALYs gained throughout the projection phase are summarized in Fig 3.

Over 10 years, we estimated the highest incremental cost in Ulaanbaatar (USD 661,313), and the lowest in Dakar (USD 375,541). For São Paulo, we estimated incremental costs of USD 483,014. The estimated ICER was much below the lower bound of the threshold in São Paulo (USD 784 per QALY gained) and Ulaanbaatar (USD 748 per QALY gained). In Dakar, assuming implementation from Q1 2019-Q4 2019 and an increase in control rates from 13% to 19%, the estimated ICER exceeds the threshold (USD 3,091 per QALY gained). Our exploratory analysis considering Q2 2018 with a control rate of 7% as baseline, yielded an ICER of USD 1,755, which

**Table 4. Estimated CHD events, strokes, deaths, and events averted following implementation of the CARDIO4Cities approach in Ulaanbaatar, Dakar, and São Paulo for a 2-year implementation phase and a 10-year projection phase.** Event projections for "No CARDIO" assume control rates consistent with baseline control rates (3% for Ulaanbaatar, 13% for Dakar, and 12% for São Paulo). "CARDIO" assumes control rates equivalent to those measured in the last quarter of CARDIO implementation (19% for Ulaanbaatar, 19% for Dakar, and 30% for São Paulo). Control rates were assumed to remain consistent throughout the projection period.

| | Ulaanbaatar (n=10,075) | | | Dakar (n=5236) | | | São Paulo (n=5844) | | |
|---|---|---|---|---|---|---|---|---|---|
| | No CARDIO | CARDIO | Averted | No CARDIO | CARDIO | Averted | No CARDIO | CARDIO | Averted |
| **Implementation phase** | | | | | | | | | |
| *CHD* | | | | | | | | | |
| Total | 175 | 157 | 18 (10.3%) | 101 | 98 | 3 (3.0%) | 158 | 139 | 19 (12.0%) |
| Per 1,000 patients | 17.4 | 15.6 | 1.8 | 19.3 | 18.7 | 0.57 | 27.0 | 23.8 | 3.3 |
| *Stroke* | | | | | | | | | |
| Total | 52 | 47 | 5 (9.6%) | 30 | 29 | 1 (3.3%) | 47 | 41 | 6 (12.8%) |
| Per 1,000 patients | 5.2 | 4.7 | 0.5 | 5.7 | 5.5 | 0.2 | 8.0 | 7.0 | 1.0 |
| *Deaths* | | | | | | | | | |
| Total | 262 | 262 | 0 | 65 | 65 | 0 | 167 | 167 | 0 |
| Per 1,000 patients | 26.0 | 26.0 | 0.0 | 12.4 | 12.4 | 0.0 | 28.6 | 28.6 | 0.0 |
| **Ten-year projection** | | | | | | | | | |
| *CHD* | | | | | | | | | |
| Total | 869 | 836 | 33 (3.8%) | 890 | 865 | 25 (2.8%) | 1,042 | 961 | 81 (7.8%) |
| Per 1,000 patients | 86.3 | 83.0 | 3.3 | 170.0 | 165.2 | 4.8 | 178.3 | 164.4 | 13.9 |
| *Stroke* | | | | | | | | | |
| Total | 261 | 247 | 14 (5.4%) | 268 | 258 | 10 (3.7%) | 313 | 282 | 31 (9.9%) |
| Per 1,000 patients | 25.9 | 24.5 | 1.4 | 51.2 | 49.3 | 1.9 | 53.6 | 48.3 | 5.3 |
| *Deaths* | | | | | | | | | |
| Total | 3,761 | 3,507 | 254 (6.8%) | 1,249 | 1,215 | 34 (2.7%) | 2,343 | 2,159 | 184 (7.9%) |
| Per 1,000 patients | 373.3 | 348.1 | 25.2 | 238.5 | 232.0 | 6.5 | 400.9 | 369.4 | 31.5 |

also fell above the uncertainty bounds for the cost-effective ICER threshold (S2 Text). We therefore projected the initiative to be strongly cost-effective within 10 years in Ulaanbaatar and São Paulo. For Dakar, cost-effectiveness could not be confirmed as data at baseline was insufficient to conduct the analysis. For São Paulo, the approach is simulated to breakeven in year 6 of the projection period, when the cumulative costs in the 'No CARDIO' scenario exceed those under the 'CARDIO' scenario. The cost-effectiveness analysis is summarized in Table 5.

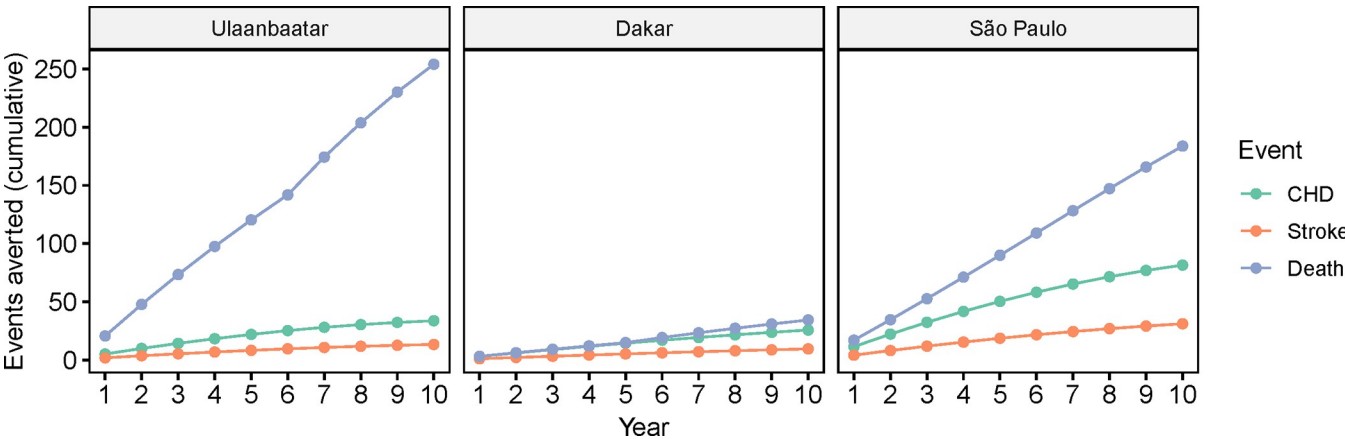

**Fig 2. Estimated cumulative number of CV events and deaths averted 10 years after CARDIO implementation.**

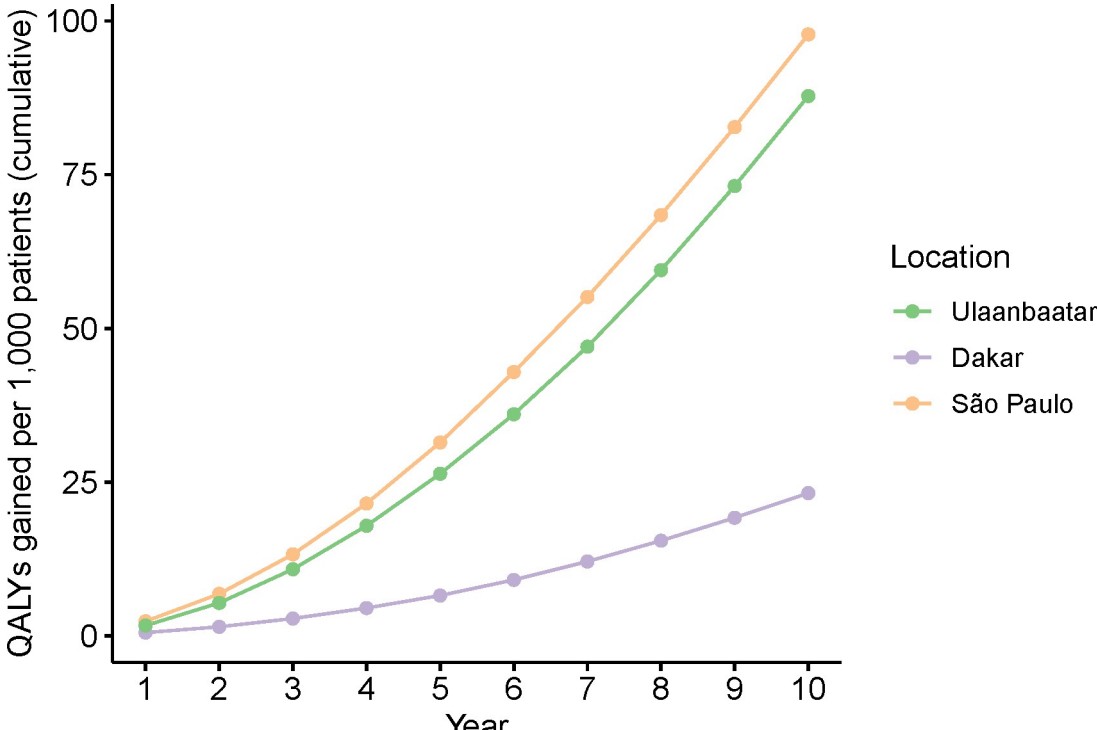

**Fig 3. Predicted cumulative number of QALYs gained per 1,000 treated patients under CARDIO compared to the 'No CARDIO' scenario over a 10-year projection phase in Ulaanbaatar, Dakar, and São Paulo.**

**Table 5. The incremental cost summarizes the difference in total costs (program cost and CV-event costs) for the No CARDIO vs CARDIO scenario.** [+]Estimates extracted from [40]. The thresholds are adjusted for PPP and in-country opportunity costs. *Estimates extracted from [43]. [$]WHO CHOICE stipulates a cost-effectiveness threshold of three times the per-capita GDP.

|  | Ulaanbaatar | Dakar | São Paulo |
|---|---|---|---|
| **Total cost savings from events averted (in USD)** | | | |
| CHD (implementation phase) | 4,944 | 3,622 | 29,249 |
| CHD (projection phase) | 8,097 | 24,715 | 99,391 |
| Stroke (implementation phase) | 6,133 | 5,072 | 38,493 |
| Stroke (projection phase) | 12,302 | 35,244 | 145,490 |
| **Costs (in USD)** | | | |
| Incremental Cost | 661,313 | 375,541 | 448,249 |
| **QALYs** | | | |
| Incremental QALYs | 884 | 121 | 572 |
| Incremental QALYs / 1,000 patients | 88 | 23 | 98 |
| **Cost-effectiveness (USD)** | | | |
| ICER (per QALY gained) | 748 | 3,091 | 784 |
| ICER threshold (per QALY gained)[+] | 1,624-4,849 | 73-1,166 | 3,210- 10,122 |
| National-level GDP per capita in 2018[*] | 4,135 | 1,458 | 9,151 |
| GDP-based ICER threshold (WHO-CHOICE)[$] | 12,405 | 4,374 | 27,453 |
| **Cost-effective** | **Yes** | **Unclear** | **Yes** |
| **Breakeven** | **No** | **No** | **Year 6** |

## Sensitivity analysis

The one-way parameter sensitivity analysis yielded similar results for Dakar and Ulaanbaatar, where the predicted ICER was most sensitive to variations in the mean population age (Fig 4). For São Paulo, ICER values were most sensitive to the assumed cost associated with hypertension in controlled patients and to the assumed risks of CV events (stroke and CHD). For São Paulo and Ulaanbaatar, all upper estimates of the sensitivity analysis fell below the lower bound of the more conservative cost-effectiveness thresholds (adjusted for PPP and opportunity costs), affirming the conclusion that the initiative will likely be cost-effective in both cities. For Dakar, cost effectiveness could never be established for the adjusted threshold but could in most cases be reached for the unadjusted threshold.

## Discussion

CVD remains the largest contributor to disease burden in LMICs with sustained increases driven by epidemiological transition, rapid urbanization, and overwhelmed health systems [44,45]. The need for impactful population-level interventions that are also cost-effective is evident. In this study, we refer to an implementation study of an urban population health initiative to improve hypertension management that implemented the CARDIO4Cities approach [28]. The beneficial effect of this approach on improving population-level hypertension control has been previously documented [29]. Here, we provide complementary evidence for the short- and long-term population-level impact of the CARDIO4Cities approach on CVD-associated morbidity and mortality and note that the approach was strongly cost-effective in Ulaanbaatar, Mongolia and São Paulo, Brazil. Our results suggest that the CARDIO4Cities approach, implemented in collaboration with local governments to strategically complement and accelerate ongoing care processes, can generate long-term population health impact after implementation periods of only one to two years.

Across the three cities, estimated reductions in CV event rates ranged from 3% to 12% over one to two years of implementation and reporting. In Ulaanbaatar, an estimated 12.8% of strokes were averted, 3.3% in Dakar, and 9.6%% in São Paulo. Simultaneously, an estimated 12% of CHD events were averted in Ulaanbaatar, 3% in Dakar, and 10.3% in São Paulo. Over the following 10 years, we predicted 2.8–7.8% of strokes, 3.7–9.9% of CHD events and 2.7–7.9% of deaths averted. Additionally, despite social, cultural, and economic differences between the cities, the CARDIO4Cities approach was predicted strongly cost-effective (ICER <20% of the lower cost-effectiveness threshold bound) in São Paulo and Ulaanbaatar. For Dakar, cost-effectiveness would be met under the WHO-CHOICE-stipulated threshold of

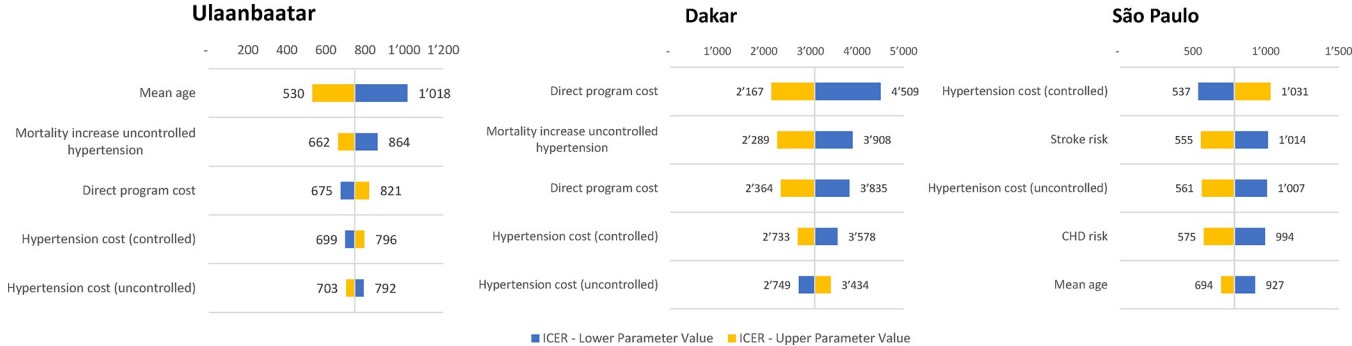

**Fig 4. One-way sensitivity analysis of cost effectiveness across three cities.** Each parameter was varied independently by 10% and ICERs calculated.

three times the per-capita GPD. However, when considering a threshold adjusted for PPP and opportunity costs, cost-effectiveness could not be established, regardless of the assumed starting quarter and control rate (see also Tables A and B in S3 Text). In Dakar, the cost-benefit ratio of CARDIO was generally less favorable than in the other cities, with high implementation costs at lower incremental health benefits (23–50 QALYs gained per 1000 patients compared to 88 in Ulaanbaatar and 99 in São Paulo). This may result on the one hand from the limited availability of chronic health services in the city at the start of the initiative, reflected in a high proportion (86.1%) of newly diagnosed hypertensive patients at baseline, as compared to 5.2% in Ulaanbaatar and 11.9% in São Paulo [29]. On the other hand, it may also be attributable to the relatively low absolute number of patients achieving control, resulting from a small number of total patients combined with only a moderate increase in control rates. In Dakar, the absolute number of patients controlled for hypertension at the end of CARDIO implementation (995 patients) was approximately half that of the other two cities (1,914 patients in Ulaanbaatar and 1,812 patients in São Paulo). Consequently, fewer events were averted. Generally, cost-effectiveness thresholds are not prescriptive. Our results should therefore not be interpreted as a counterindication to the implementation of CARDIO in a Dakar-like setting. Additionally, as our model predictions are based on control rates, they do not consider the absolute number of patients that would have gone undiagnosed, untreated, and therefore uncontrolled in the absence of the initiative. Rather than strictly adhering to thresholds, decision-makers must decide themselves if the population health benefit of an intervention is worth its cost. If doubts arise, an enhanced risk evaluation and detailed planning should be conducted to reduce costs and enhance the benefits of a health intervention.

Identifying scalable, cost-efficient, and swiftly implementable solutions is especially important for LMICs, where elevated pressure on the health system and limited resources contribute to poor CV health outcomes. Across many LMICs, population-level interventions targeting primary and secondary prevention of CVD have been shown to yield substantial reductions in CVDs and CVD-mortality [46,47] and many of these interventions have been shown to lead to costs savings for the respective health systems [46]. A population health approach that combines a robust, yet flexible, locally adaptable approach provides an attractive solution. With activities within and outside the health system, CARDIO contributes to the comprehensive strengthening of CVD management, relying on optimizing early detection of CV risks and standardizing frontline hypertension management. Overall, the CARDIO4Cities approach has potential to simplify and improve CV population health management, while narrowing health inequities and can be replicated or adapted in diverse contexts or for other health conditions. To ensure that the improved CV population health management reaches all patients, future initiatives should aim to understand the impact of social and broader determinants on health equity.

To achieve impact and enable replication, the CARDIO4Cities approach requires strong political will and interest from authorities to invest in their health system and health corps, as well as a readiness of health workers and their managers to adopt the required changes of standardization and data integration. When tailored to the local needs and integrated into existing health workflows, the CARDIO4Cities approach can both improve population CV health and strengthen health system performance through encouraging best practices and data-driven decision making. This systematic integration of real-time data into decision making is important to improve health planning and allocation of scarce resources to those interventions that can have the largest impact on the greatest number of people.

Our modelling study estimates possible real-world outcomes but is limited by modelling assumptions and data shortcomings. Limitations of the health outcomes data and collection methodology have been reported previously [29]. BP measurement practices vary globally. As

the first step of the clinical cascade of care, the coverage of early detection can greatly influence the impact of any population health effort. To minimize this variation, CARDIO4Cities included the development of standard guidelines for early detection of hypertension in each city in collaboration with local authorities and medical societies. These guidelines were integrated into official care protocols. They adhered to standards of the WHO and the International Society of Hypertension (ISH) and included requirements for validated measurement devices and methods, e.g., checking for behavior that could influence BP (recent physical activity, smoking, alcohol or food consumption), ensuring the patient rests for 3–5 minutes before measurement and is seated during measurement, and taking a minimum of three measurements per appointment, at 1 minute intervals, with an arm cuff at heart level. The initiative was fully integrated into ongoing health system processes and relied on local ownership. Therefore, adherence to guidelines was not independently audited by the initiative, limiting its ability to control for potentially confounding external influences. For example, the procurement of BP measurement devices within standard local operating procedures was in part supported by the initiative (e.g., Screening corners in São Paulo, see S1 Table). Validation and maintenance of the devices was addressed in official care protocols (e.g., in São Paulo: guidance for annual validation and calibration of measuring cuffs). Yet, responsibility for execution remained with the local health system. Similarly, adherence to care guidelines was emphasized through repeated trainings of health care professionals, and regular quality reviews were established, but all processes were locally driven. We do not foresee a systematic or intentional deviation from the guidelines in any of the participating cities, but differences in the implementation of clinical practice are possible and could bias our results. To maximize coverage of early detection, active BP measurement was implemented complementary to routine practice. In all three cities, all patients entering primary health units were offered to have their BP measured and were subsequently followed up depending on BP levels. Further, community outreach events were conducted to integrate asymptomatic healthy, prospective hypertension patients into care. For this analysis, we consider patients as "hypertensive" if their measured BP was above the hypertension threshold in the local clinical protocols. More stringent thresholds in Ulaanbaatar (130/80 mmHg) than in the other cities (140/90 mm Hg) mean that patients categorized as hypertensive in Ulaanbaatar may have been considered "normal" in the other cities, leading to a relative overestimation of the hypertensive population in our analysis. Additionally, due to the comprehensiveness of the approach and its implementation in a real-world setting without strict randomization and with continued refinement, it is not possible to attribute the impact of the approach to specific interventions. With the available data, the components of the approach can only be evaluated jointly, and the "No CARDIO" scenario used as a comparator in this analysis relies on projections from baseline values. Due to its integrated implementation within ongoing care processes, the CARDIO4Cities group was also not directly involved in the generation of the data. Individual-level data or data on potentially confounding variables such as adherence to treatment or co-morbidities were not available, leading us to base our model on population-level characteristics. Additionally, the data in Dakar was insufficient to conduct the desired analysis, and additional approximations had to be made. Variables for the cost-effectiveness analysis had to be indirectly derived or interpolated, while the analysis converts changes in hypertension control rates into estimated event rates and QALYs averted. Although we followed standard practices, the conversions used are approximate and the accuracy of projected CHD events and strokes should be confirmed as longitudinal data is collected. Additionally, local cost for CHD events and strokes were unavailable for Senegal and Mongolia, and approximations based on PPP conversions from the São Paulo data were used. Nevertheless, we anticipate that our conclusions on the long-term impact and cost-effectiveness of the CARDIO4Cities approach are robust enough to

overcome the data shortcomings. Conservative estimates and assumptions were used throughout our analysis (e.g., assuming no further improvements in BP control rates after the end of CARDIO implementation) and a sensitivity analysis conducted to account for parameter uncertainties. As estimated event rates were extrapolated from hypertension control categories (controlled vs. uncontrolled), the model did not account for continuous risk lowering through decreased BP within one control category, potentially underestimating the effect size. For São Paulo, the collaborative roadmap design for implementation with local authorities and clinics already started in December 2017. In this study, refer to Q4 2018, the time of intervention roll-out, as the baseline to evaluate the cost-effectiveness of the solutions. This definition contributes to additional potential downward bias in the estimated effect size reported here, as the conversations and design thinking processes prior to implementation are likely to have already positively impacted care processes. Lastly, although we assessed the impact and cost-effectiveness in cities across three continents, the generalizability of findings will be further assessed while the approach is scaled to more locations. For example, CV risk estimation by use of the Framingham Risk Score is not fully validated in all countries. With these considerations, we emphasize that our results are a modelling extrapolation on the cost-effectiveness of the CARDIO4Cities approach, based on its early results. Our results are encouraging and provide evidence to the impact and cost-effectiveness of the approach. Yet, real-world data on cardiovascular event rates or health spendings is needed to confirm the robustness of our modelling results.

## Conclusions

Overall, our results demonstrate that reducing CVD morbidity and mortality by improving hypertension control at population level is both clinically impactful and broadly economically cost-effective once enough patients are involved. Most importantly, due to the direct relationship between blood pressure and the risk of CV events, improvements in population health can be generated rapidly, even after short intervention periods. The estimated impact of the CARDIO4Cities approach further reflects the potential of public private sector collaborations and simplified approaches that hold patients and health workers at the center, where data is integrated to continuously monitor progress and guide the population health interventions.

## Supporting information

**S1 File. Full model.**
(XLSM)

**S1 Table. Summary of CARDIO activities by city.**
(DOCX)

**S1 Text. Modelling methodology.** Table A in S1 Text: Assumed CVD risk by sex and CVD risk category. Table B in S1 Text: Patient distribution across sex and CVD risk category for each location and scenario. Table C in S1 Text: Initial blood pressure of patients who achieved BP control under the 'CARDIO' and 'NO CARDIO' scenarios. Table D in S1 Text. Model Assumptions and parameters.
(DOCX)

**S2 Text. Cost assumptions.** Table A in S2 Text: Phases of work in the three cities. Table B in S2 Text: Assumed implementation cost for the CARDIO approach in Ulaanbaatar, Dakar, and São Paulo.
(DOCX)

**S3 Text. Alternative simulation for Dakar.** Table A in S3 Text: Estimated CHD events, strokes, deaths, and events averted following implementation of the CARDIO approach in, Dakar under alternative assumptions for a 2-year implementation phase and a 10-year projection phase. Event projections for "No CARDIO" assume control rates consistent with baseline control rates (7%). "CARDIO" assumes control rates equivalent to those measured in the last quarter of CARDIO implementation (19%). Control rates were assumed to remain consistent throughout the projection period. Table B in S3 Text: The incremental cost summarizes the difference in total costs (program cost and CV-event costs) for No CARDIO vs CARDIO case. +Estimates extracted from [36]. The thresholds are adjusted for PPP and in-country opportunity costs. *Estimates extracted from [38]. $WHO CHOICE stipulates a cost-effectiveness threshold of three times the per-capita GDP.
(DOCX)

## Acknowledgments

We specifically thank the national authorities from Mongolia and Senegal and municipal authorities from São Paulo, Brazil, Ulaanbaatar, Mongolia and Dakar, Senegal for leading and owning the urban population health initiative; all health providers, managers, nurses, doctors, community agents and other health professionals who participated in the initiatives co-creation and roll out; and all patients who participated in the implementation. We are grateful to all local partners and stakeholders who contributed to the initiative, with a special thanks to the teams of the Onom Foundation and MPHPA in Mongolia, IntraHealth and PATH in Senegal, and Instituto Tellus and Iqvia in Brazil. We are grateful to the UMANE Foundation for the strategic partnership of the initiative in São Paulo, Brazil.

## Disclosure

The views expressed in this manuscript are those of the authors and do not represent the views of the American Heart Association.

## Author Contributions

**Conceptualization:** Sarah Des Rosiers, Johannes Boch, Gautam Partha, Abhinav Srivasatava, Yara C. Baxter, Tumurbaatar Luvsansambuu, Ann Aerts.

**Data curation:** Gautam Partha, Abhinav Srivasatava, Joseph Barboza, Enkhtuya Byambasuren, Renato W. de Oliveira.

**Formal analysis:** Theresa Reiker, Gautam Partha, Abhinav Srivasatava.

**Investigation:** Theresa Reiker, Sarah Des Rosiers, Gautam Partha, Abhinav Srivasatava, Yara C. Baxter.

**Methodology:** Theresa Reiker, Sarah Des Rosiers, Johannes Boch, Gautam Partha, Lakshmi Venkitachalam, Adela Santana, Abhinav Srivasatava.

**Software:** Abhinav Srivasatava.

**Supervision:** Theresa Reiker, Sarah Des Rosiers, Johannes Boch, Lakshmi Venkitachalam, Abhinav Srivasatava, Ann Aerts.

**Validation:** Theresa Reiker, Sarah Des Rosiers, Johannes Boch, Abhinav Srivasatava, Yara C. Baxter, Karina Mauro Dib, Naranjargal Dashdorj, Malick Anne, Renato W. de Oliveira,

Mariana Silveira, Jose M. E. Ferrer, Louise Morgan, Olivia Jones, Tumurbaatar Luvsansambuu, Luiz Aparecido Bortolotto, Luciano Drager, Alvaro Avezum, Ann Aerts.

**Visualization:** Gautam Partha.

**Writing – original draft:** Theresa Reiker, Sarah Des Rosiers, Gautam Partha, Abhinav Srivasatava.

**Writing – review & editing:** Theresa Reiker, Johannes Boch, Lakshmi Venkitachalam, Adela Santana, Joseph Barboza, Enkhtuya Byambasuren, Yara C. Baxter, Karina Mauro Dib, Naranjargal Dashdorj, Malick Anne, Renato W. de Oliveira, Mariana Silveira, Jose M. E. Ferrer, Louise Morgan, Olivia Jones, Tumurbaatar Luvsansambuu, Luiz Aparecido Bortolotto, Luciano Drager, Alvaro Avezum, Ann Aerts.

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
