## [Decision Letter · Decision Letter 0]

26 Oct 2022

PGPH-D-22-01401

Population health impact and economic evaluation of a multisectoral initiative to improve urban hypertension management

Dear Dr. Reiker,

Thank you for submitting your manuscript to PLOS Global Public Health. After careful consideration, we feel that it has merit but does not fully meet PLOS Global Public Health’s publication criteria as it currently stands. Therefore, we invite you to submit a revised version of the manuscript that addresses the points raised during the review process.

We look forward to receiving your revised manuscript.

Kind regards,

Jasper Tromp

Academic Editor

Journal Requirements:

a. Please clarify all sources of funding (financial or material support) for your study. List the grants (with grant number) or organizations (with url) that supported your study, including funding received from your institution. 

b. State the initials, alongside each funding source, of each author to receive each grant.

c. State what role the funders took in the study. If the funders had no role in your study, please state: “The funders had no role in study design, data collection and analysis, decision to publish, or preparation of the manuscript.”

d. If any authors received a salary from any of your funders, please state which authors and which funders.

2. We ask that a manuscript source file is provided at Revision. Please upload your manuscript file as a .doc, .docx, .rtf or .tex.

Additional Editor Comments (if provided):

Reviewers' comments:

Reviewer's Responses to Questions

**Comments to the Author**

1. Does this manuscript meet PLOS Global Public Health’s publication criteria? Is the manuscript technically sound, and do the data support the conclusions? The manuscript must describe methodologically and ethically rigorous research with conclusions that are appropriately drawn based on the data presented.

Reviewer #1: Yes

Reviewer #2: Partly

2. Has the statistical analysis been performed appropriately and rigorously?

Reviewer #1: Yes

Reviewer #2: No

3. Have the authors made all data underlying the findings in their manuscript fully available (please refer to the Data Availability Statement at the start of the manuscript PDF file)?

Reviewer #1: No

Reviewer #2: No

4. Is the manuscript presented in an intelligible fashion and written in standard English?

Reviewer #1: Yes

Reviewer #2: Yes

5. Review Comments to the Author

Reviewer #1: This is well-written article concerning the economic evaluation over specific urban populations’ health, resulting from a hypertension management program under the CARDIO4Cities approach. In this sense, this is rather the second, economic evaluation part of the survey and not strictly speaking a scientific paper concerning hypertension management, which would demand much more information, related references and cross-discussion. The fact that it is the economic evaluation of the CARDIO4Cities approach has to be clearly stated to the title and thorough the text.

As this study is the expansion of a previously published work in order to support the existing approach, there is no discussion concerning the screening criteria for hypertension, which not only are different between the three cities, but are also different throughout the world.

Moreover, major discrepancy, somehow stated, concerns the measurement procedure of pressure. Which are the measurement protocols? It is widely known that even if they do exist they are not always followed. On the other hand, when the measurements are performed with electronic pressure meters an overestimation is common. What about some procedures to ensure the proper function of the devices.

I know that it is not common in your field to apply uncertainty estimations, nevertheless when all these bias parameters are present sensitivity analysis is not enough. Submitted reference [26] indicates many limitations. Some of them are: i) it was not possible to assess the specific impact of interventions tackling underlying determinants of hypertension, ii) a strictly randomized setup (e.g. a randomized control trial) was not possible, iii) the CARDIO4Cities group was not involved in the generation of the data, iv) there was a lack of in-country consensus on the methodology quality and usefulness for clinical practice.

You have applied a lot of advanced statistical tools to get into the economical aspects of CARDIO4Cities approach, but what about the grounds that it has been set?

Additionally, major hypertension sources are the living and working conditions, somehow covered by the urbanization notion, but they are much wider. Part of them are the psycho-social issues that have also great importance in the working environment. There are major European organizations dealing with these issues (e.g. EUROFOUND, EU-OSHA).

What about GDPR issues of the medical records?

Finally, I don’t see many non-medication interventions and mainly what about the screening of these non-medication acts. For sure the whole project is based on the intention to protect the health of urban LMICs population, but do they get the same attention as the high income countries?

Check out Abstract’s spelling; there are some capitalizations that create bad impression.

Reviewer #2: Background. It is necessary to explore the reasons for choosing the title of this research and prove it with the latest data for the last 5 years

Method. A more comprehensive analysis is needed, namely bivariate and multivariate analysis.

Discussion. need to be discussed further. not just copying the results

conclusion. no conclusions were found in the articles written

bibliography. not according to the guide

6. PLOS authors have the option to publish the peer review history of their article (what does this mean?). If published, this will include your full peer review and any attached files.

**Do you want your identity to be public for this peer review?** For information about this choice, including consent withdrawal, please see our Privacy Policy.

Reviewer #1: **Yes: **George A Gourzoulidis

Reviewer #2: No

---

## [Decision Letter · Decision Letter 1]

14 Feb 2023

PGPH-D-22-01401R1

Population health impact and economic evaluation of the CARDIO4Cities approach to improve urban hypertension management

Dear Dr. Reiker,

Thank you for submitting your manuscript to PLOS Global Public Health. After careful consideration, we feel that it has merit but does not fully meet PLOS Global Public Health’s publication criteria as it currently stands. Therefore, we invite you to submit a revised version of the manuscript that addresses the points raised during the review process.

Your manuscript has been evaluated by one of the previous reviewers. They are satisfied with most of the changes you have made, but have provided a few further comments for you to address. Please ensure you address in particular detail the reviewer's comment regarding an extra statement of limitation, and please also carry out a careful copyedit of your manuscript text.

We look forward to receiving your revised manuscript.

Kind regards,

Hugh Cowley

Staff Editor

Journal Requirements:

2. We have noticed that you have uploaded Supporting Information files, but you have not included a list of legends. Please add a full list of legends for your Supporting Information files after the references list.

Additional Editor Comments (if provided):

Reviewers' comments:

Reviewer's Responses to Questions

**Comments to the Author**

1. If the authors have adequately addressed your comments raised in a previous round of review and you feel that this manuscript is now acceptable for publication, you may indicate that here to bypass the “Comments to the Author” section, enter your conflict of interest statement in the “Confidential to Editor” section, and submit your "Accept" recommendation.

Reviewer #1: (No Response)

2. Does this manuscript meet PLOS Global Public Health’s publication criteria? Is the manuscript technically sound, and do the data support the conclusions? The manuscript must describe methodologically and ethically rigorous research with conclusions that are appropriately drawn based on the data presented.

Reviewer #1: Yes

3. Has the statistical analysis been performed appropriately and rigorously?

Reviewer #1: Yes

4. Have the authors made all data underlying the findings in their manuscript fully available (please refer to the Data Availability Statement at the start of the manuscript PDF file)?

Reviewer #1: Yes

5. Is the manuscript presented in an intelligible fashion and written in standard English?

Reviewer #1: Yes

6. Review Comments to the Author

Reviewer #1: This is a much more improved version of the manuscript.

The majority of the requests have been met clarifying the scope, the limitations and the alternative approaches.

Nevertheless, the reported screening guidelines that are reported to be developed by CARDIO4Cities, according to WHO standards, in order to access BP are not included in the submission. In this sense, an extra limitation concerning procedures to ensure the proper function of the screening devices should be added (e.g. specifications, calibration).

Moreover in lines 89-90 the notion ‘including non-medication interventions’ after ‘primary care level’ should be included.

Some linguistic and spelling check will benefit the text.

7. PLOS authors have the option to publish the peer review history of their article (what does this mean?). If published, this will include your full peer review and any attached files.

**Do you want your identity to be public for this peer review?** For information about this choice, including consent withdrawal, please see our Privacy Policy.

Reviewer #1: **Yes: **George A Gourzoulidis

---

## [Editor Report · Decision Letter 2]

28 Feb 2023

Population health impact and economic evaluation of the CARDIO4Cities approach to improve urban hypertension management

PGPH-D-22-01401R2

Dear Dr Reiker,

We are pleased to inform you that your manuscript 'Population health impact and economic evaluation of the CARDIO4Cities approach to improve urban hypertension management' has been provisionally accepted for publication in PLOS Global Public Health.

Best regards,

Julia Robinson

Executive Editor